# The Influence of Subjective Perceptions and the Efficacy of Objective Evaluation in Soccer School Players’ Classification: A Cross-Sectional Study

**DOI:** 10.3390/children10050767

**Published:** 2023-04-23

**Authors:** Federico Abate Daga, Franco Veglio, Gian Maria Cherasco, Samuel Agostino

**Affiliations:** 1Department of Clinical and Biological Sciences, University of Turin, Orbassano, 10043 Turin, Italy; 2Department of Medical Sciences, University of Turin, 10126 Turin, Italy

**Keywords:** talent identification, prepubertal football, field-test battery, emotional perceptions

## Abstract

Objectives: The first objective was to investigate the influence of subjective perceptions and the efficacy of objective evaluation regarding the classification of soccer school players by their level of performance. The second objective was to advise on accurate collocation according to objective assessment of players’ performance. Methods: An objective evaluation of the players’ motor performance abilities and coaches’ subjective perceptions of the players’ levels of performance was conducted with 34 young football players (U11) from Torino FC soccer school. The players were allocated to three groups based on their perceived performance level at the start of the season. The players were evaluated using a field test battery, and team coaches provided subjective ratings of the players’ abilities. Results: MANOVA showed significant differences between the playing levels (F = 2.185, *p* < 0.05; partial η2 = 0.34) in the 10 × 5 shuttle run, 20 m sprint test (objective evaluations), heading, understanding of the game, positioning on the field, speed and agility (subjective perception) (F = 1.951, *p* < 0.05; partial η2 = 0.43). A discriminant analysis of the field test scores revealed that 76.5% of players were correctly categorised in one of the three performance-level groups. However, the first group (the best players) had the lowest predictive accuracy rate (58.3%). By comparison, the second group (the average players) had a much higher predictive accuracy rate (83.3%), and the third (the weakest players) had the highest (90.0%). Conclusions: These findings support the role of objective performance evaluation in categorising players of different skill in soccer schools.

## 1. Introduction

Football is one of the most popular sports in the world [1] and is practised worldwide during childhood and adolescence. The International Federation of Association Football (FIFA) estimated that approximately 265 million people play football worldwide, and almost 22 million of these are registered players under 18, considered “youth” [2]. This vast number of players is usually involved in youth teams belonging to soccer schools or academies at every level. As a result, elite, sub-elite and amateur clubs have to organise their squads and group the players according to their performance level every season. Grouping players according to their level (current and potential) of performance is one of the most challenging tasks for a soccer school or academy coordinator. It can be considered as difficult as talent identification when scouting players [3,4]. However, while today the talent identification process is well-studied [5,6] and methodologically defined by the scientific literature [7], the act of group formation according to players’ level of performance is still a subjective process. Among field experts, a frequent habit is considering players’ ranking by performance as a part of the coaches’ job [8,9], usually accomplished during training or matches and based on the perception of their performance and ability. A recent study [10] aimed to identify coaches’ criteria for classifying players by their skills and abilities, determining that technical, tactical, and mental factors are most often considered in talent identification. However, establishing that a player is technically, tactically, or mentally skilled depends wholly on the coach’s feelings instead of rather than objective data. The so-called “Halo Effect” strongly affects subjective perceptions, particularly when positive [11]. Therefore, players’# judgement may be strongly influenced by positive feelings about skills or abilities that may only partially represent a player’s performance level. Thus, the factors that influence coaches’ judgement need to be investigated. According to the literature [12], one of these may be related to the players’ levels of so-called “fundamental motor skills.” Coaches seem to perceive those demonstrating high levels of fundamental motor skills as better players. Conversely, other studies indicate that biological maturation, fitness status, and physical dimensions are the discriminant factors involved in the selection, as well as minutes played in elite youth football (soccer) [13,14]. Under those circumstances, estimating how effectively coaches discriminate between players during group formation is complex. Moreover, considering that first-choice players usually compete in the most challenging games, grouping players by their skills and abilities plays a vital role in the process of talent development in football. Thus, this study investigates the influence of subjective perceptions and the efficacy of objective evaluation in soccer school players’ classification. Secondly, this research aims to provide an accurate classification according to the objective measurement of players’ performance.

## 2. Materials and Methods

### 2.1. Study Design and Ethical Considerations

This cross-sectional study was approved by the Institutional Research Bioethics Committee of the University of Turin (0258948). The children and coaches in this study belonged to an Italian professional club soccer school. Participation was voluntary, and parents consented to their children’s involvement by signing an informed consent statement. Therefore, children without parental approval were excluded from the study. The study was conducted in May 2022 (the end of the 2021–2022 season) on the U11 players of Torino FC (Italian Serie A).

At the beginning of the 2021–2022 season, the players were divided into three squads according to coaches’ and soccer school coordinators’ perceptions of their performance level. In addition, each group received a head and two assistant coaches, while the goalkeeper and fitness coach were the same for all groups. Children had to meet the following inclusion criteria to be part of this study. Firstly, they had to be outfield players regularly involved in football training (i.e., 2–3 training sessions and one match per week) and part of the club since at least the previous season. Finally, they must not have been injured for a long time. Conversely, children would be excluded if they played as goalkeepers, did not train regularly (one session or fewer per week) or were recruited during the ongoing season.

Furthermore, they were excluded in case of severe injuries and if their BMI was above the 95th percentile for their age (i.e., U11:22.06 kg/m^2^). Children’s objective evaluations were assessed using a field test battery with the U11 fitness coach. On the other hand, subjective evaluations were performed by head coaches using a questionnaire. Each coach had to pass judgement via a score from one to five for each questionnaire item. This process was undertaken for squad players and the remaining two groups (Figure 1). 

### 2.2. Participants

Forty-seven prepubertal footballers and three coaches were recruited for this study. Participants belonged to the U11 of a professional club soccer school. At the beginning of the season, children were divided into three groups according to coaches’ and coordinators’ perceptions of their level of performance. The first group comprised the club’s best players of this age. The second group was formed of average players. Finally, the third group contained the weakest players. At the beginning of the season, each coach received their group. Then, assignments were made considering the coach’s curriculum, the past season’s scores, and the target reached.

### 2.3. Procedures

The field test battery was performed during a training week in May 2022. Measurements were conducted on the same artificial playing pitch to avoid an irregular surface that might affect the outcomes. The testing sessions were conducted at the beginning of the training schedule, on a sunny day representative of typical training conditions. The children’s training took place on Mondays, Wednesdays, and Fridays, each lasting two hours, from 5 p.m. to 7 p.m. Each program involved a 15-min warm-up and 15 min of “free match” at the beginning and end of the training. The schedule’s central part was composed of three main topics of 30 min each: motor and fitness development, skills and abilities development, and tactics. During the experimental period, objective evaluations were performed at the beginning of the training schedule, replacing the motor and fitness development part. 

The children arrived at the training pitch, changed their clothes, and went directly to the fitness and conditioning coach to warm up and prepare for the testing session. During the physical trials, the fitness coach led the schedule while the researchers set up the proper test, recorded data and checked the validity of the performance. 

The first training session involved anthropometrics, flexibility, and neuromuscular mechanisms (lower limb power using the standing long jump test and pure speed using the 20 m sprint test). Then, on the second training day, the ability to lead the ball at maximum speed on different distances with a change in direction (technical and coordinative evaluation with the Shuttle Dribble Test) and maintain speed under continuous effort with (10 × 5 shuttle run test) were assessed. Finally, changes in aerobic fitness were evaluated using the Mini Cooper test in the third session of the week; 48-h rests should be observed between sessions.

Additionally, coaches had to fill out a questionnaire to obtain a subjective evaluation of the players’ soccer-specific qualities.

Before starting the testing session, the children performed standardised warm-up exercises of 5 min of 2 vs. 2 Small-Sided Games (S.S.G.s), stretching, dynamic stretching, and trial runs of each test. Then, the players were split into groups of 4–6, not necessarily belonging to a single group classification. Thus, players from different groups could be tested together. This procedure guaranteed an adequate number of players per testing station, improving their attention and helping them focus on the test performance. Before each test, children received clear verbal instructions and demonstrations. Each test was completed within one or two attempts, depending on the test protocol. The best one was taken for further analysis in the case of two trials. 

While the children were involved in the testing session, the head coach completed the questionnaire to assess the quality of soccer players. The contemporaneity of objective evaluation and subjective perception assessment was utilised to optimise time and training management. The questionnaire was filled in away from the children (at the bench or changing room) to avoid outside influence on the testing scores. 

### 2.4. Assessment of Anthropometric Status

Body mass was measured to the nearest 0.1 kg (Rowenta BS1060, Erbach, Germany), with the participants wearing their football equipment except for shoes and shin guards. Standing height was calculated using a wall stadiometer with a precision of 0.01 m and a 60–210 cm range (Lanzoni D01602 H, Bologna, Italy). BMI was calculated using a Microsoft Excel sheet, where the BMI formula was previously inserted (BMI = Body weight/(height*height)).

Skeletal age was not measured. Even if it has been demonstrated that skeletal age can be considered the best measure of biological variability [15], it does not significantly affect motor coordination in children up to 10 years old [16].

### 2.5. Assessment of Motor Performance Abilities

In a previous study, motor performance abilities were assessed with a field-test battery used by Abate Daga and his colleagues in an earlier study [17]. This protocol ensured the evaluation of the most significant motor performance abilities required in football. In addition, all tests were part of the Eurofit Physical Fitness Test Battery (designed to measure physical fitness in school-age children) or were set explicitly for children and previously used in several studies. For this reason, the present field-test battery was chosen and utilised in this research.

### 2.6. Sit and Reach Test

The sit and reach Test (S.R.) followed the same protocol described by Daga and colleagues [18]. To perform the S.R., the subject sat on the floor with head, back and hips against a wall, knees straight, legs together, and soles of the feet positioned flat against the S.R. box (height: 30 cm, width: 50 cm, depth: 51 cm). The starting point of the bar was represented by the 0 cm mark on the measuring scale. The bar runs along the S.R. box’s upper face, and the bar with the measuring scale is 80 cm long. The place where the feet were in contact with the box was 30 cm from the starting point of the bar. Before starting the test, the player extended their arms with palms facing down and the index fingers in contact. Then, to perform the test, the player had to slowly bend forward while elongating their knees and having their hands slide on the measuring scale. The researcher registered the score and ensured that the subject’s heels remained in the box and the knees were fully extended during the performance. Each player was only given one attempt. (Enabling players to repeat attempts during a testing session can transform it into flexibility training.) In case of test failure, the player was discharged and recalled to a new testing session on another training day.

### 2.7. Standing Long Jump Test

The objective of the test was to determine the distance an athlete can jump while standing still. To perform the test, the athlete stood behind a white line on the court with their feet separated at shoulder width. Then, they jumped using both feet while swinging their arms and bending their knees for propulsion. The goal was to land on both feet without falling backwards while jumping as far as possible. Finally, a researcher recorded the result by measuring from the white line to the nearest heel. Each player had two attempts, and the best score was taken for further analysis.

### 2.8. 20-mM Sprint Test

The test involves running a maximum sprint over 20 m, with the time recorded. The player started from stationary, with one foot in front of the other. The front foot was behind the starting line, delimited by the white back line of the football pitch. Two gates of photocells (Microgate Witty, Bolzano, Italy) were displaced at the starting and the finish line. Thus, the player could start running when they felt ready to go. When the run started, the tester provided hints of maximising speed (such as keeping low and driving hard with the arms and legs) and encouraged them to continue running hard past the finish line. This encouraged all children to do their best. Each player performed two trials; the best score was taken for further analysis. 

### 2.9. Shuttle Dribble Test

The shuttle dribble test, originally developed to assess field hockey performance, involves carrying a ball while completing 30-m shuttle sprints at various distances. However, it has also been validated for use with soccer players ([19]). 

The test required the placement of two pairs of photocells (Microgate Witty, Bolzano, Italy) on the starting line, spaced 1 m apart. After each change, the player performed 180° direction changes at 5, 6, 10, and 9 m, returning to the starting line. The test was completed at maximum speed while carrying the ball and within a 2-m passageway. The test was conducted twice, and the best score was recorded for evaluation.

### 2.10. 10 × 5 Shuttle Run

In this test, players must maintain their maximum running speed while completing 10 consecutive 5-m shuttle sprints, returning to the starting line after each change in direction. A professional manual chronometer (HS-3 V-1 RET Casio, Japan), capable of sampling at 0.01 s, was used to record the time from the start signal. The test was repeated once, and the time was recorded for further analysis.

### 2.11. Mini Cooper Test

This test consists of walking or running continuously for 6 min. Specifically [20], children were instructed to walk or run around a 9 × 18 m rectangle as quickly as possible for 6 min, with the option to walk if needed. The distance covered during the test was manually recorded on an evaluation sheet. The children were divided into smaller groups of a maximum of 10 players before the trial to ensure accurate monitoring and distance registration.

#### The Questionnaire for the Assessment of Football Player Quality by the Coach

The questionnaire was utilised for the first time in the study of Jukic and colleagues [12] and partially modified to fit this research better. The questionnaire consisted of nine elements focused on technical, tactical, physical, and psychological characteristics: (1) passing and control of the ball; (2) leading the ball; (3) running with the ball; (4) the finishing technique at the goal; (5) heading; (6) understanding of the game and position on the field; (7) attitude towards the coach and training sessions; (8) competitiveness and enthusiasm before a match; and (9) speed and agility [12]. Players were evaluated with a score from 1 to 5 (1 = feeble; 2 = somewhat flawed; 3 = average performance, 4 = good; 5 = excellent).

### 2.12. Statistical Analysis

All data were analysed using SPSS, version 19.0 (SPSS Inc., Chicago, IL, USA). 

Descriptive statistics ((mean and standard deviation (S.D.)) were used to present participants’ demographic data.

As noted, the motor performance abilities assessment provided five scores (standing long jump, 20-m sprint, shuttle dribble test, 10 × 5 shuttle run and Mini Cooper test) in different domains (meters and seconds). At the same time, the Questionnaire for the Assessment of Football Player Quality by the Coach reported scores from nine items ((1) passing and control of the ball; (2) leading the ball; (3) running with the ball; (4) the finishing technique at the goal; (5) heading; (6) understanding of the game and position on the field; (7) attitude towards the coach and training sessions; (8) competitiveness and enthusiasm before a match; and (9) speed and agility [12]. All these outcomes were considered dependent variables, and z-scores were used to compare them. Multivariate analysis of variance (MANOVAs) and Bonferroni post hoc tests were conducted to detect any significant difference between the variables. Partial eta-squared (ηp2) was used to analyse the magnitude of effects using cut-off scores of small (0.01), moderate (0.06) and strong (0.14) effects [21].

Secondly, linear discriminant analysis (LDA) was employed to analyse the data. This involved simultaneously inserting the independent variables into the equation using a standard procedure.

Additionally, the LDA was utilised to create discriminant functions by linearly combining the measured variables. This method helped classify participants into playing levels based on multiple motor performance variables. Again, a homogenous variance and multivariate normal within-group distribution were assumed for this analysis. 

Two discriminant functions were developed for the entire sample and then used to classify the participants of the same study group into the established categories. A varimax rotation was performed on the significant function to control the standardised canonical coefficients of the discriminant function. The discriminant or predictor variables were the sit and reach test, standing long jump (S.L.J.), 20 m sprint, shuttle–dribble test, 10 × 5 shuttle run, and Mini Cooper test. Pearson’s correlation coefficients were calculated to indicate the agreement between the arbitrary group composition and objectively measured performance characteristics. Significance was assumed at *p* < 0.05.

## 3. Results

Forty-seven young Italian soccer players were recruited for this study. However, 13 were excluded because they did not meet the inclusion criteria. In particular, six were goalkeepers, three were recruited during the ongoing season, and four did not train regularly (one or fewer sessions per week). Therefore, only 34 children were eligible for this study, and their data were considered for further analysis. The mean weight was 39.64 ± 5.24 kg, the mean height was 144.74 ± 5.76 cm and the mean BMI was 18.91 ± 2.08 kg/m^2^. The sample belonged to the U11 Italian Serie A club Torino FC soccer school. Participants’ anthropometric characteristics are given in Table 1.

Before introducing MANOVA and linear discriminant analysis results, it is essential to declare that Mahalanobis distances detected no outliers in all variables. Moreover, Pearson’s correlations were used to detect any possible strong correlation among variables. No strong correlations were identified among variables in either objective or subjective evaluations (Table 2 and Table 3). Furthermore, a multiple regression analysis was conducted to identify collinearity between objective and subjective variables. The variance inflation factor (V.I.F.) demonstrated no multicollinearity in all variables (V.I.F. = 1.48). Finally, in the Box Tests, the observed covariance matrices of the dependent variables were equal across groups.

To analyse the role of the objective evaluation (measured by a field-test battery) and subjective perception of the player’s performance level (measured with a specific questionnaire) on group levelling, field-test outcomes and questionnaire results were analysed using two MANOVAs.

First of all, there was a significant effect of group level of performance (first, second, third) F (12, 52) = 2.185, *p* < 0.05; ” ’Wilk’s Λ = 0.442, partial η2 = 0.34. In addition, the results showed a significant main effect in the 10 × 5 shuttle run test F (2, 31) = 6.596, *p* < 0.05 partial η2 = 0.20. Furthermore, a Bonferroni post hoc test showed significant differences between the first and the third group (*p* < 0.05), while no differences were reported between the first and second group (*p* = 0.121), or the second and third group (*p* = 1.000). (Table 1) A significant effect was also found in 20 m sprint F (2, 31) = 5.612, *p* < 0.01 partial η2 = 0.26, and a Bonferroni post hoc test revealed significant differences between the first and the third group (*p* < 0.05) and the second and the third group (*p* < 0.05). No differences were detected between the first and the second group (*p* = 1.000) (Table 1). 

Considering the subjective evaluation, there was a significant effect of perceived level of performance (first, second, third) F (2, 31) = 1.951, *p* < 0.05; ” ’Wilk’s Λ = 0.322, partial η2 = 0.43. 

In particular, the results showed a significant main effect in heading F (2, 31) = 8.202, *p* < 0.001 partial η2 = 0.35, understanding of the game and position on the field F (2, 31) = 3.525, *p* < 0.05 partial η2 = 0.19, and speed and agility F (2, 31) = 5.823, *p* < 0.01 partial η2 = 0.27. 

Moreover, a Bonferroni post hoc test revealed a significant difference between the first and the third group in heading (*p* < 0.001), understanding of the game, position on the field (*p* < 0.05) and speed and agility (*p* < 0.01) (Table 1).

Secondly, a trend of significance was observed in heading between the second and third groups (*p* = 0.059). Finally, no significant differences were detected between the first and second groups in heading, understanding of the game, position on the field, and speed and agility.

The MANOVA was followed by a discriminant analysis, revealing two discriminant functions. A summary of the discriminant variables for each skill level group is given in Table 4. The first function explained 56.8% of the variance, (canonical *R*^2^ = 0.59), whereas the second explained 43.2% (canonical *R*^2^ = 0.419).

In combination, these discriminant functions significantly differentiated the three groups, Λ = 0.44, χ 2 (12) = 9.51, *p* = 0.025. Still, removing the first function indicated that the second function did not significantly differentiate between the groups, Λ = 0.70, χ 2 (5) = 9.98, *p* = 0.076. Table 4 summarises the standardised coefficients of the discriminant functions f1 and f2. 

The relative values of those coefficients, according to the standardised canonical discriminant function coefficients, reveal that the most relevant predictors are as follows: for the first function, the 20-m sprint (r = 0.859), the S.L.J. (r = 0.592), and the shuttle–dribble test (r = 0.388); and for the second function, the shuttle run (r = −0.667), sit and reach (r = 0.499) and Mini Cooper (r = 0.435). However, a structured matrix rotation (varimax rotation) was necessary to understand better the discriminant function’s standardised coefficient (Table 5). Therefore, after the varimax rotation process, the prediction of function 1 was confirmed by the following standardised canonical coefficients: 20 m sprint (r = 0.927), the S.L.J. (r = 0.575), and the shuttle–dribble test (r = 0.485). Thus, these results showed that the 20-m sprint, the S.L.J., and the shuttle–dribble test were the most relevant predictors for categorising the players in the present study. 

The localisations of the participants according to the values obtained from the first discriminant function are shown in Figure 2. In addition, the group centroid for the three skill-level groups is also reported. The horizontal and vertical separation between group centroids reflects the discriminatory power of f1 compared to f2, which was nonsignificant. 

The discriminant functions served as a model to propose players’ performance level assignments following the values of their respective predictor variables. Table 6 compares the actual classification of all young footballers with the one predicted by the model. Notably, 76.5% of players were correctly categorised in one of the three performance-level groups. Therefore, misclassification was 23.5%. However, the discriminant functions were found to possess a higher predictive accuracy for the third group (90.0%) and the second group (83.3%).

Conversely, only 58.3% of players were correctly allocated for the first group.

## 4. Discussion

The primary purpose of the present study was to investigate the influence of subjective perceptions and the efficacy of objective evaluation in soccer school players’ classification. Secondly, this research aimed to produce an accurate classification by objectively measuring players’ performance. 

The present study’s findings show that objective evaluation can provide more valuable information about players’ level of performance than coaches’ subjective perceptions. Therefore, assessing the level of performance in selected prepubertal footballers only by subjective perceptions may limit the talent identification and development process. 

For example, the post hoc test of the 20-m sprint test detected a significant difference between all groups (the first and the second on the third) (*p* < 0.05). In contrast, the item “speed and agility” on the questionnaire for coaches’ subjective perceptions only indicated a difference between the first and third groups (*p* < 0.01). Thus, an objective test can identify the differences in the level of performance of prepuberal players.

This may confirm that coaches’ subjective perceptions are less or not properly efficient when the level of performance is similar. In addition, coaches may be affected by conscious or unconscious philosophical and cognitive biases ([9]). Thus, there might be a bias in performance evaluation [22]. Moreover, sprinting capacity has been recognised as a discriminant characteristic of football performance [23,24]. Considering this, classifying players according to this objective parameter may result in more efficient grouping and, consequently, more proficient football formation for soccer school players.

Furthermore, the 10 × 5 shuttle run test identified a significant difference between the first and third groups (*p* < 0.05). Sprinting, high-speed running, and nonlinear running still characterise football matches involving youth. Shuttle sprints reproduce the conditions of football matches, where continuous direction changes are required ([25]). Thus, an objective evaluation of this parameter may be helpful for grouping. Conversely, this characteristic is difficult to assess during matches and training, and no item in the questionnaire could identify it. Unfortunately, the scarcity of research among selected prepubertal footballers makes comparing these findings difficult.

Furthermore, the novelty of this study focuses on established football players and their allocation to hierarchical groups. Most of the literature investigates selected versus nonselected players. Thus, some similarities were observed in the study by Jukic and colleagues [12]. They observed better scores in locomotory skills from the first group than the second, but this difference is insignificant. The present study’s findings partially agree with the analysis of Jukic and colleagues (2019) because they detected some nonsignificant changes in their objective evaluation battery involving sprint ability tests.

Conversely, this study noted a significant difference between the groups in the 20-m sprint test. In particular, the first and second groups performed better than the third. However, no difference was detected between the first and second groups. This may be due to the dimensions and structure of the sample. This research involved three levels of performance instead of two and 34 players instead of 23.

Furthermore, these findings agree with Jukic and colleagues [12], reporting no differences between the first and second groups. Once selected, the first and second groups probably exhibit very similar performance levels, so their classification should be determined in another way. Therefore, adding a third level of performance is necessary to manifest a significant difference in motor performance abilities. This agrees with the existing literature, which reports the overperformance of selected players compared with lower-level or nonselected players in various team sports [26,27]. 

As it concerns subjective evaluation, this study found a significant difference in heading, understanding of the game, position on the field, and speed and agility. Again, the only possible comparison is with the study of Jukic and colleagues [12], and, once more, this study partially agrees with them. This research found the same significant difference in heading, understanding of the game and position on the field that the cited authors reported. However, this study showed a significant difference between the first and third groups. In the cited research, it was between the first and second groups. In addition, they did not detect a significant difference in speed and agility, while this study reported it. This can be explained by the bigger sample size (34 vs. 23), the presence of one more group, and the similar level of performance between the first and second groups in this study compared to the third.

Conversely, no differences were identified in passing and controlling the ball, leading the ball, running with the ball, the finishing technique at the goal, and the attitude towards the coach and training sessions. However, Jukic and colleagues [12] registered differences among these variables. The presence of a third coach and different reference scenarios (analytical exercise versus small-sided games or other situations) may lead to this difference. Additional research is necessary to understand the influence of analytical activity and dynamic environments (i.e., training matches or small-sided games) on coaches’ subjective perceptions.

The second aim of this study was to provide an accurate classification system according to objective evaluation based on the field-test battery. The discriminant analysis showed that 76.5% of the original group was correctly classified. However, almost 25% of the players were not correctly allocated, generating possible bias. In addition, the discriminant function offered high predictive accuracy for the second and third groups, while the first group showed the highest inaccuracy. Only 58.3% of the players belonging to the first group were correctly allocated with arbitrary classification.

Conversely, the second and third groups had 83.3% and 90% of the players correctly assigned, respectively. These findings suggest that random typecasting based on subjective feelings is much more efficient at grouping lower-level players than higher ones. Thus, coaches and staff believe in identifying talent, but without the support of objective data, they are more successful at scouting and categorising lower-level players. Future research should investigate this behaviour by helping coaches to better understand the mechanisms governing the subjective perception of players’ level of performance. 

Finally, this study has its limitations. First, these data refer to a limited number of children of a specific age category belonging to one professional club. The outcomes might change if all age categories of a soccer school or more than one club were involved in the measurements. Secondly, subjective perceptions might be unconsciously influenced by feelings and emotions derived from the past or the incoming match and generate a bias in personal evaluation. However, the critical strength of this study lies in investigating the classification of selected players of a professional soccer school and underlining the importance of objective evaluation to better identify future talent. For this reason, this research may provide helpful, practical applications for experts and professionals in youth football training. First, coaches and soccer school managers should avoid relying on personal feelings and subjective perceptions of performance when grouping players.

Conversely, a field-test battery to evaluate motor performance abilities may be a helpful tool for classification. Secondly, dynamic categorisation should be adopted during the regular season. Players’ motor performance abilities and football skill improvement should be assessed using objective evaluations, and team rosters should be reformatted according to the young footballers’ new performance level.

## 5. Conclusions

In conclusion, the present findings emphasise the role of objective evaluation in determining players’ levels according to motor performance abilities and football skills. Furthermore, the results revealed that accurate testing of motor performance abilities could be an appropriate way to help coaches achieve more precise evaluation using only their subjective perceptions of prepubertal footballers’ level of performance. In particular, the 20-m sprint test and 10 × 5 shuttle run can reveal physical performance differences among groups of selected players from the same age category. Finally, an arbitrary categorisation based on subjective feelings is more efficient at identifying lower-level players than the best ones. Thus, an objective evaluation based on field-test batteries may be essential to mitigate this bias and make the groups more homogeneous to maximise the football schedule, coaching, and growing talent.

## Figures and Tables

**Figure 1 children-10-00767-f001:**
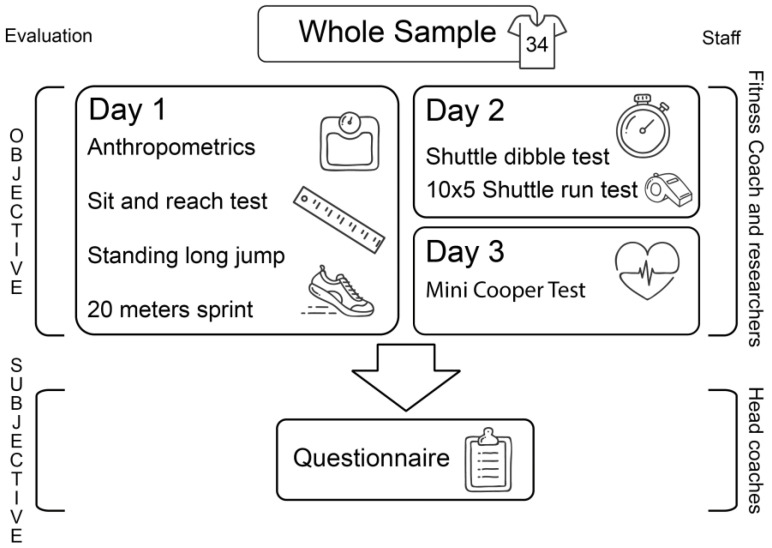
Graphical representation of the experimental design.

**Figure 2 children-10-00767-f002:**
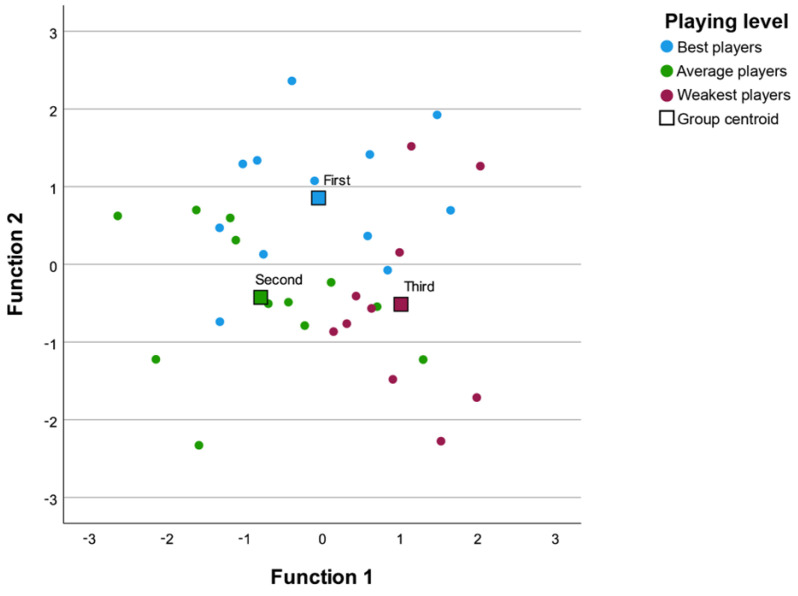
Topographical map of players’ level of performance considering the discriminant functions. Centroids represent the mean variate scores for each group. (First group: light blue, Second group: green, Third group: light purple).

**Table 1 children-10-00767-t001:** Playing level differences in anthropometry, objective evaluation (physical tests) and coaches’ subjective perception (questionnaire items).

	Playing Level			MANOVA		
	Best Players *n* = 12	Average Players *n* = 12	Weakest Players *n* = 10	Partial Eta Squared (ηp2)	F	*p*
Age	11 ± 0.20	11.1 ± 0.21	11.1 ± 0.24	--	--	--
Anthropometry						
Body mass (kg)	38.74 ± 3.33	40.11 ± 5.8	40.16 ± 6.65	0.02	0.262	0.771
Stature (cm)	144.21 ± 4.62	144.75 ± 5.04	145.35 ± 8.02	0.01	0.101	0.904
BMI	18.67 ± 1.84	19.12 ± 2.28	18.97 ± 2.28	0.01	0.136	0.873
Physical tests						
Shuttle run (s)	17.79 ± 0.67	18.38 ± 0.66	18.54 ± 0.73 *	0.2	3872	<0.05
Mini Cooper (m)	1168.75 ± 63.18	1120.83 ± 82.46	1123 ± 93.75	0.08	1342	0.276
Sit and reach (cm)	30.88 ± 4.57	27.21 ± 6.89	26.4 ± 4.82	0.12	2095	0.14
20-m sprint (s)	3.38 ± 0.17	3.37 ± 0.16 **	3.58 ± 0.15 *	0.27	5612	<0.01
Shuttle dribble (s)	9.48 ± 0.46	9.4 ± 0.42	9.9 ± 0.9	0.11	1984	0.155
SLJ (cm)	164.92 ± 12.14	155.17 ± 16.87	161.9 ± 12.73	0.09	1484	0.242
Subjective evaluations						
Passing and control of the ball	3.38 ± 0.57	2.75 ± 0.54	2.75 ± 1.21	0.13	2358	0.111
Leading the ball	3.38 ± 0.48	3.25 ± 0.45	3.05 ± 1.12	0.04	0.561	0.576
Running with the ball	3.25 ± 0.58	3.08 ± 0.56	2.6 ± 0.94	0.14	2503	0.098
The finishing technique at the goal	3.42 ± 0.63	3.17 ± 0.62	2.6 ± 1.02	0.17	3234	0.053
Heading	2.71 ± 0.62	2.33 ± 0.44	1.75 ± 0.59 *	0.35	8202	<0.001
Understanding of the game and their position on the field	3.38 ± 0.57	2.83 ± 0.54	2.6 ± 0.99 *	0.18	3525	<0.05
Attitude towards the coach and training sessions	4.33 ± 0.65	4.54 ± 0.45	4.2 ± 0.89	0.04	0.729	0.49
Competitive character and enthusiasm before a match	3.79 ± 0.81	3.62 ± 0.64	3.25 ± 1.3	0.06	0.95	0.398
Speed and agility	3.58 ± 0.87	2.96 ± 0.69	2.35 ± 0.97 *	0.27	5823	<0.01

Note: * significant differences (*p* < 0.05) concerning group “Best players”; ** significant differences (*p* < 0.05) to group “Weakest players”.

**Table 2 children-10-00767-t002:** Pearson’s correlation of the objective evaluation. Significance was identified with *p* < 0.05.

	Shuttle Run (s)	Mini Cooper (m)	Sit and Reach (cm)	20 m Sprint (s)	Shuttle Drible (s)	S.L.J. (cm)
Shuttle run (s)	--					
Mini Cooper (m)	−0.151	--				
Sit and reach (cm)	−0.137	0.336	--			
20 m sprint (s)	0.515 **	−0.206	−0.129	--		
Shuttle dribble (s)	0.296	−0.277	−0.194	0.334	--	
SLJ (cm)	−0.303	0.462 **	0.385 *	−0.184	−0.185	--

** Correlation is significant at the 0.01 level (2-tailed). * Correlation is significant at the 0.05 level (2-tailed).

**Table 3 children-10-00767-t003:** Pearson’s correlation of the subjective evaluation. Significance was identified with *p* < 0.05.

	Passing and Control of the Ball	Leading the Ball	Running with the Ball	The Finishing Technique at the Goal	Heading	Understanding of the Game and Their Position on the Field	Attitude towards the Coach and Training Sessions	Competitiveness and Enthusiasm before a Match	Speed and Agility
Passing and control of the ball	--								
Leading the ball	0.805 **	--							
Running with the ball	0.723 **	0.894 **	--						
The finishing technique at the goal	0.608 **	0.621 **	0.705 **	--					
Heading	0.454 **	0.508 **	0.642 **	0.653 **	--				
Understanding of the game and position on the field	0.772 **	0.706 **	0.655 **	0.447 **	0.565 **	--			
Attitude towards the coach and training sessions	0.143	0.164	0.109	−0.160	0.142	0.391 *	--		
Competitiveness and enthusiasm before a match	0.637 **	0.696 **	0.794 **	0.573 **	0.540 **	0.581 **	0.175	--	
Speed and agility	0.633 **	0.711 **	0.779 **	0.592 **	0.700 **	0.651 **	0.177	0.678 **	--

** Correlation is significant at the 0.01 level (2-tailed). * Correlation is significant at the 0.05 level (2-tailed).

**Table 4 children-10-00767-t004:** Discriminant function structure coefficients and tests of statistical significance.

	Function	
Variable	1	2
Shuttle run (s)	−0.035	−0.917
Mini cooper (m)	−0.014	0.426
Sit and reach (cm)	−0.346	0.455
20-m sprint (s)	0.859	0.353
Shuttle dribble (s)	0.388	0.364
SLJ (cm)	0.592	0.046
Wilks’ lambda	0.442	0.704
Chi-square	23.274	9.983
*P*	0.025	0.076
Eigenvalue	0.594	0.419
% of variance	58.6	41.4
Canonical correlation	0.611	0.544
**Functions at group centroids**		
Best players	−0.322	0.792
Average players	−0.615	−0.659
Weakest players	1124	−0.16

Discriminant function structure coefficients and tests of statistical significance.

**Table 5 children-10-00767-t005:** Varimax rotation of the variables on the significant function.

Rotated Standardised Canonical Discriminant Function Coefficients		
	Function	
	1	2
20 m sprint (s)	0.927 *	0.057
SLJ (cm)	0.575 *	−0.147
Shuttle dribble (s)	0.485 *	0.219
Shuttle run (s)	−0.329	−0.857 *
Sit and reach (cm)	−0.181	0.542 *
Mini Cooper (m)	0.124	0.408 *

The absolute size of correlation within the function is used to order variables. * Largest absolute coefficient of the variable among the discriminant functions

**Table 6 children-10-00767-t006:** Classification matrix for players’ actual and predicted playing levels according to discriminant functions.

	Predicted Group		
Actual Group	Team 1	Team 2	Team 3
Team 1 (*n* = 12)	58.3% (*n* = 7)	25% (*n* = 3)	16.7% (*n* = 2)
Team 2 (*n* = 12)	0% (*n* = 0)	83.3% (*n* = 10)	16.7% (*n* = 2)
Team 3 (*n* = 10)	10% (*n* = 1)	0% (*n* = 0)	90% (*n* = 9)

## Data Availability

The data presented in this study are available on request from the corresponding author. The data are not publicly available due to club owner restrictions.

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
