# Peer review of "The Influence of Subjective Perceptions and the Efficacy of Objective Evaluation in Soccer School Players’ Classification: A Cross-Sectional Study"

_children, 2023, doi:10.3390/children10050767_

Round 1

Reviewer 1 Report

This article is easy to read and provides meaningful knowledge about how's the effect of objective evaluation on soccer school players' discrimination.

The selection criteria for the test battery should be explained.

There needs to be more explanation regarding the results attained in this study from a practical point of view. 

The results should be better explained, considering their practical implications for the field.

Reviewer 2 Report

With regard to manuscript: Police verso, police recto. The influence of subjective perception and the efficacy of objective evaluation in soccer school players' discrimination: a cross-sectional study, for consideration in Children. This is a very interesting manuscript, but  I have comments and questions that I have detailed below.

·  Title: Police verso, police recto must be removed or explained. There is no other mention of this sentence.

·  Abstract: details of the 3 groups were not mentioned clearly in the abstract. Please provide more information about this.

·  The introduction is very well written.

·  Line 97: Coaches were assigned to one of these three groups at the beginning of the season. How coaches were randomized into groups. There was some discontent from the coach when he was assigned to the weakest players (i.e. group 3).

·  Line 105: Maximum speed on different distances with the change of direction. This should be taken as AGILITY (but not resistance to speed). Additionally, didactics would improve with the inclusion a figure (some scheme drawn by the authors) explaining their timeline of testing days and study design. I would like you to use your creativity in something like below.

·  Line 107: Changes in aerobic fitness. Here, it must be mentioned what tests were considered in the evaluation of aerobic fitness (I presume the Mini Cooper Test). Moreover, other components related to the alatic anaerobic (20 meters sprint test) and latic anaerobic performance (x5 Shuttle run) were assessed and must be valued.

·  Line 113: the players were split into smaller groups. How many athletes ?

·  Line 120: While the children were involved in the testing session, the head coach completed the questionnaire to assess the quality of soccer players. This phrase worried me. It seems that the decision of head coaches was influenced by the U11 fitness coach, which may have conveyed positive or negative feelings to the head coaches. The evaluation of head coach is not free from the dirty influence of testing session. Field-tests should be carry out without any communication between the head coach and U11 fitness coach. I would like this point of view to be more in-depth.

·  The authors should explain why their findings aggregates to the existing knowledge. Some comparisons with the study of Jukic and colleagues were made, but the novelty of the present study could be more highlighted.

·  Table 1: Instead of using the nomenclature “effect size”, I suggest using the eta square (η2). This is necessary to avoid confusions since that effect size also be used as Cohen's d. I only ask the authors to carefully review if you are using Eta-squared (η2) and partial eta-squared (ηp2) since they are different. See for example: Timothy R. Levine, Craig R. Hullett, Eta Squared, Partial Eta Squared, and Misreporting of Effect Size in Communication Research, Human Communication Research, Volume 28, Issue 4, October 2002, Pages 612–625. Mordkoff JT.  A Simple Method for Removing Bias From a Popular Measure of Standardized Effect Size: Adjusted Partial Eta Squared. Journal of Evidence-Based Complementary & Alternative Medicine. September 2019:271-286.

·  Table 1: Arrows (> <) in Post-Hoc seems inappropriate, especially for Shuttle run (sec) and 20mt sprint (sec). I suggest removing the last column of table 1 and insert symbols within table: Statistical analysis: # significant differences (p<0.05) in relation to group 1. * significant differences (p<0.05) in relation to group 2.

·  Table 1: It is not didactic for the reader the discrimination of playing level in number 1, 2 and 3. The groups should be named as best players, average players and weakest players.

·  The results should be more explored as by adding a table with Pearson's correlations between data from objective evaluations and data from subjective evaluations.

·  Discriminant analysis (table 2 and 3) should be more easily accessible, particularly in the mathematical details and software steps. Supplementary Methods/Results are intended to be an addition to the main manuscript. I would like to visualize more details.

·  Line 313: The present study's findings show that objective evaluation is more potent than coaches' subjective perception in assessing the player's level of performance. This phrase can be improved towards the appreciation of physical tests. Since the beginning of time, the physical performance was always well characterized by tests or challenges. It's curious the arrogance of coaches in guessing (only visually) who will do better in sport. Many talents in the world were lost due to personal taste of coaches neglecting physical tests. For this reason, I agree with authors in line 391. This is a very nice study emphasizing the importance of objective evaluation to identify future talents better.

·  Describe the experimental conditions in more detail. About data collection, give more details on place of evaluations, time of day, instructions, time in each individual evaluation and others minor things.

·  Line 340: Conversely, this study noted a significant difference between the first, third, and second and third groups in the 20-meter sprint test. This should make it easy to the reader.

Round 2

Reviewer 2 Report

Overall the manuscript has greatly improved and the authors should be commended for the thoroughness of their revisions.